

# The long-term dynamics of serum antibodies against SARS-CoV-2

Graziele Fonseca de Sousa[1,*], Thuany da Silva Nogueira[1,*], Lana Soares de Sales[1], Fernanda Ferreira Maissner[1], Odara Araújo de Oliveira[1], Hellade Lopes Rangel[1], Daniele das Graças dos Santos[1], Rodrigo Nunes-da-Fonseca[1], Jackson de Souza-Menezes[1], Jose Luciano Nepomuceno-Silva[1], Flávia Borges Mury[1], Raquel de Souza Gestinari[1], Amilcar Tanuri[2], Orlando da Costa Ferreira Jr[2] and Cintia Monteiro-de-Barros[1]

[1] Instituto de Biodiversidade e Sustentabilidade, Universidade Federal do Rio de Janeiro, Macaé, Rio de Janeiro, Brazil
[2] Laboratório de Virologia Molecular, Departamento de Genética, Instituto de Biologia, Universidade Federal do Rio de Janeiro, Rio de Janeiro, Brazil
* These authors contributed equally to this work.

Corresponding authors
Orlando da Costa Ferreira Jr,
orlandocfj@gmail.com
Cintia Monteiro-de-Barros,
cintiabarrosmacae@gmail.com

## ABSTRACT

**Objective:** To analyze the long-term dynamics of antibodies against SARS-CoV-2 and understand the impact of age, gender, and viral load on patients' immunological response.

**Methods:** Serum samples were obtained from 231 COVID-19 positive patients from Macaé, in Rio de Janeiro state, in Brazil, from June 2020 until January 2021. The production of IgA, IgM, IgG, and IgE against S glycoprotein was analyzed using the S-UFRJ assay, taking into account the age, gender, and viral load.

**Results:** Analysis of antibody production over 7 months revealed that IgA positivity gradually decreased after the first month. Additionally, the highest percentage of IgM positivity occurred in the first month (97% of patients), and declined after this period, while IgG positivity remained homogeneous for all 7 months. The same analysis for IgE revealed that almost all samples were negative. The comparison of antibody production between genders showed no significant difference. Regarding the age factor and antibody production, patients aged ≥60 years produced almost twice more IgA than younger ones (17–39 years old). Finally, a relationship between viral load and antibody production was observed only for older patients.

**Conclusions:** Our work provides an overview of long-term production of antibodies against SARS-CoV-2, suggesting prolonged production of IgA and IgM antibodies for 3 months and continued IgG production for over 7 months. In addition, it identified a correlation between viral load and IgM titers in the older group and, finally, different IgA production between the age groups.

## INTRODUCTION

COVID-19 is caused by severe acute respiratory syndrome coronavirus 2 (SARS-CoV-2), whose first cases were reported in Wuhan, in Hubei Province, in China, in December 2019, subsequently spreading all over the world. The COVID-19 pandemic caused considerable mortality worldwide, with moderate and severe cases. However, approximately 80% of all cases reported are from mildly symptomatic patients (*Wu & McGoogan, 2020*). The death rate in Brazil was very high, at 119.9 per 100 thousand individuals, and the mean age of non-survivors was over 70 (*Sanchez et al., 2021*). However, some municipalities, such Macaé, had the one of lower mortality indices of the northeastern of Rio de Janeiro in Brazil in which, the first COVID-19 case was recorded on March 24th, 2020 (*Prefeitura Municipal de Macaé, n.d.*).

SARS-CoV-2 is a positive-sense, single-stranded RNA virus with four structural proteins: spike (S) glycoprotein, envelope (E) protein, membrane (M) protein nucleocapsid (N) protein and it also has 16 nonstructural proteins (nsp1–16) (*Chen, Liu & Guo, 2020*). Nonetheless, the S glycoprotein is critical in the process of SARS-CoV-2 invading host cells, binding to them *via* the angiotensin-converting-enzyme-2 (ACE-2) receptor and mediating membrane fusion and virus entry (*Ou et al., 2020*). The S glycoprotein also plays a crucial role in elucidating the immune response during disease progression, and is known to represent a major target for neutralizing antibodies, thus making it a key antigen for the development of specific and sensitive tests to evaluate the antibody response to SARS-CoV-2 (*To et al., 2020*; *Alvim et al., 2022*). Previous study demonstrated the importance of S glycoprotein as a target of immunological response, in which the IgG produced is directed to the SARS-CoV-2 receptor-binding domain (RBD) accounts for half of S protein-induced antibody responses (*Gao et al., 2020*). Many types of enzyme-linked immunosorbent assays (ELISAs) based on S glycoprotein have been previously developed, showing minimal cross-reactivity with sera against circulating "common cold" coronaviruses (but with some cross-reactivity with SARS and MERS-COV antisera) (*Alvim et al., 2022*). These serological methods have been used to investigate the poorly understood immunological response of the host and have been used to better comprehend how the virus generates an inflammatory stage, and how it induces serological antibody production (*Taefehshokr et al., 2020*).

However, despite having many studies on the production of antibodies, few are carried out analyzing the long-term dynamic of antibody production against SARS-CoV-2, some of them, have revealed that, high levels of antibodies could be associated with the abnormal inflammatory response, disease severity and with the presence of comorbidities. It is known that the outcome of severe SARS-CoV-2 infection varies widely and, the age factor is important since the majority of younger patients experience mild disease, while older patients display severe cases (*Brodin, 2020*). Some studies have presented evidence that antibody response may be different in children, adolescents, and adults, which potentially influences the clinical manifestations (*Weisberg et al., 2020*). However, the causes of

differential severity between ages remain unclear, presumably due to differences in the elicited immune responses (*Feitosa et al., 2021*). Furthermore, gender is an important factor to be analyzed. Males are over-represented among patients with severe disease (*Wu & McGoogan, 2020*; *Feitosa et al., 2021*). Finally, studies focusing on viral kinetics and antibody responses during the period of infection, as well as on the monitoring of COVID-19 patients for longer periods, are limited in the literature (*To et al., 2020*; *Takahashi et al., 2020*). Therefore, to better understand the immunological response to SARS-CoV-2, numerous studies have been performing, and strategies have been establishing aiming to prevent the further spread of COVID-19 and also could contribute to the development of new therapeutic approaches (*Long et al., 2020*; *Okba et al., 2020*; *Yang et al., 2021*).

Then, in order to gain a better understanding of the host immune response, the IgA, IgM, IgG, and IgE production was monitored in the serum for 7 months in mildly symptomatic patients, using the S-UFRJ assay against S glycoprotein. The patients were randomly selected based in a polymerase chain reaction (PCR)-confirmed COVID-19 infection and were grouped corresponding to how long after the first COVID-19 symptoms the serum was obtained (from first to seventh month). Additionally, the samples were analyzed taking account of the age and separated in three groups (younger, adults and older), gender (men and women), and the viral load correlating this measurement with semi-quantitative levels of IgA, IgM, IgG and IgE.

## METHODOLOGY

### Inclusion and exclusion criteria and sample collection

Serum samples were obtained from 231 positive volunteers (RT-qPCR) to COVID-19 which, were mildly symptomatic, then, the samples were grouped taking into account how long after the first COVID-19 symptoms the serum was obtained. The 231 serum samples covered approximately 7 months, from June 2020 to January 2021 ($n$ = 34; 17; 45; 22; 16; 72; 25, respectively to first until seventh month). The inclusion criteria were a RT-qPCR positive diagnosis of COVID-19 and acceptance of a term of consent received written to use the information informed from the study participants. The exclusion criteria were a negative COVID-19 test, lack of willingness or ability to provide the informed consent, lack of an appropriate legal guardian or representative, or other medical contraindications for donating blood and nasopharyngeal samples. Pregnant women, children, and adolescents under 17 years old were also excluded.

Nasopharyngeal swabs (infused in 2 mL of Hanks' balanced salt solution) and blood samples were collected in the Trial Center for COVID-19 patients (CTC) and the real-time polymerase chain reaction (RT-qPCR) (*Prefeitura Municipal de Macaé, n.d.*) and ELISA test with AC-purified S protein (S-UFRJ ELISA) (*Gao et al., 2020*) were performed at the NUPEM-UFRJ Institute and the acceptance of a term of consent for the use of

information. The study was approved by the Comitê de Ética em Pesquisa (Research Ethics Committee, Brazilian Ministry of Health: approval number 32868720.4.0000.5699). All research was performed in accordance with the relevant guidelines and regulations.

## RT-qPCR and viral load estimation

For viral RNA extraction, magnetic beads (Magmax Magnetic Kit; Thermo Fisher Scientific, Waltham, MA, USA) were used following the manufacturer's instructions. RT-qPCR reactions for the identification of SARS-CoV-2 positive samples were performed using a TaqMan™ approach, as previously described by Corman (*Corman et al., 2020*) or the CDC (*Centers for Disease Control, 2021*) protocols. RT-qPCR tests were considered positive when two regions of SARS-CoV-2 genomes were amplified. All RT-qPCR assays were performed in a QuantStudio™ 3 Real-Time PCR System (Applied Biosystems, Waltham, MA, USA).

Positivity for SARS-CoV-2 infection was determined by RT-qPCR of nasopharyngeal swabs using the Allplex™ 2019-nCoV Assay kit (Seegene Inc., Seoul, South Korea), with modifications (Supplemental Material). Viral loads were estimated based on a standard curve with serially-diluted plasmid-carrying cloned viral genes, as described by *Chung et al. (2021)*, with modifications, including normalization with endogenous RNAse P RNA (Figs. S1A, S1B).

## S-UFRJ ELISA for anti-S IgM, IgG, IgE, and IgA detection

The S-UFRJ ELISA was performed using the trimeric spike protein of SARS-COV-2 produced in stable recombinant HEK293 cells as described in *Alvim et al. (2022)* and it was provided by the Cell Culture Engineering Laboratory—LECC from the Universidade Federal do Rio de Janeiro—UFRJ. The secondary antibodies concentration used was 1:200 goat anti-human IgM (Fc) HRP-labeled antibody (#MFCD00162459; Sigma, St. Louis, MO, USA), anti-human IgG (Fc) HRP-labeled antibody (#SAB3701282; Sigma, St. Louis, MO, USA), anti-human IgE (Fc) HRP-labeled antibody (#A18793; Invitrogen, Waltham, MA, USA), and anti-human IgA (Fc) (#PA1-74495; Invitrogen, Waltham, MA, USA), incubated for 1 h at room temperature. The reaction was read at 450 nm in a Thermo Scientific Multiskan Sky microplate reader. The results were expressed either as optical density units (OD) or as the ratio of sample OD/cut-off value. The pre pandemic healthy donors (blood samples before 2019) were selected as negative controls. Then, the cut-off was made by means of O.D. of the samples plus three times the O.D. Standard deviation (*Alvim et al., 2022*).

## Statistical analysis

The entire data analysis was performed with the GraphPad Prism 8 (GraphPad Software Inc., San Diego, CA, USA) software. The data were expressed as mean ± standard error of the mean. The normal distribution was verified by the Kolmogorov-Smirnov test. The Mann–Whitney test was used for comparison between two groups and the Kruskal–Wallis test was used for comparison of more than two groups, followed by Dunn's *post hoc* test. To assess the relationship between viral load and antibody production by
**Table 1 Characteristics of mildly symptomatic SARS-CoV-2 patients.** Percentages are shown for each category with respective counts in parentheses. Select COVID-19 risk factors were scored by a clinical infectious disease physician.

| Characteristic | Patients No./Total No. | | | |
| --- | --- | --- | --- | --- |
| | Total (N = 231) | Adolescents and young adults (17–39 years old, n = 99) | Adults (40–59 years old, n = 100) | Seniors ( >60 years old, n = 32) |
| **Sex** | | | | |
| Women | 128/231 (54.9%) | 60/99 (58.8%) | 60/99 (58.8%) | 17/32 (53.1%) |
| Men | 105/231 (45.1%) | 42/99 (41.2%) | 42/99 (41.2%) | 15/32 (46.9%) |
| **Age** | | | | |
| Median (range) | 52.5 (17–89) | 28 (17-39) | 49.5 (40–59) | 74.5 (60–89) |
| **Comorbidities** | | | | |
| Bronchitis | 3/231 (1.3%) | 2/99 (1.96%) | 1/100 (1.0%) | – |
| Women | 3/3 (100%) | 2/99 (1.96%) | 1/100 (1.0%) | – |
| Men | – | – | – | – |
| Chronic cardiac disease | 37/231 (16%) | 4/99 (3.9%) | 24/100 (24%) | 9/32 (28.1%) |
| Women | 21/37 (56.8%) | 3/99 (3%) | 13/100 (13%) | 5/32 (15.2%) |
| Men | 16/37 (43.2%) | 1/99 (1%) | 11/100 (11%) | 4/32 (12.5%) |
| Diabetes | 9/231 (3.9%) | – | 6/100 (6%) | 3/32 (9.4%) |
| Women | 6/9 (66.7%) | – | 4/100 (4%) | 2/32 (6.3%) |
| Men | 3/9 (33.3%) | – | 2/100 (2%) | 1/32 (3.1%) |
| Obesity | 62/231 (26.8%) | 18/99 (18.2%) | 37/100 (37%) | 7/32 (21.9%) |
| Women | 32/62 (51.6%) | 10/99 (10%) | 17/100 (17%) | 5/32 (15.6%) |
| Men | 30/62 (48.4%) | 8/99 (8%) | 20/100 (20%) | 2/32 (6.3%) |

linear regression, the Spearman's test was used. Variance in the groups were analyzed by coefficient of variation (CV). Differences were considered to be significant when $P < 0.05$.

# RESULTS

## Overview of features

The median age of the SARS-CoV-2 patients was 52.5 years old, and 54.9% were women while 45.1% were men. The chronic diseases characterizing them were as follows: 1.3% asthmatic bronchitis, 18.6% chronic cardiac disease, 4.8% diabetes, and 32.5% obesity (Table 1).

### Serological analysis of IgA, IgM, IgG, and IgE against SARS-CoV-2

To understand the long-term response of antibodies against the entire spike glycoprotein, serological tests were performed with the serum of mildly symptomatic patients in the post disease period and analyzed semi-quantitatively for IgA, IgM, IgG, and IgE positivity. Antibody production was monitored every month for the different patients, grouped taking into account the exact period of the time after the first symptoms that the serum sample was taken, covering a total of 7 months. The samples with OD values under the cut-off were considered as not producing immunoglobulins.

Almost all the patients were IgA positive for SARS-CoV-2, but a small fraction, though positive in the RT-qPCR, were IgA negative (11.4%) in the first month. The IgA positivity among the patients decreased in the following months, showing a percentage of 64.7% in the second month, and gradually declining until reaching 53.2% in the seventh month (1.38 nm/0.40; 0.72 nm/0.50; 0.68 nm/0.65; 0.77 nm/0.55; 0.53 nm/0.44; 0.68 nm/0.50; 0.73 nm/0.52, respectively to first until seventh month; mean/coefficient of variance). The highest OD values were observed in the samples from the first month and a decrease in these values was observed until the seventh month after the first onset of symptoms. However, minor group of patients are positive to IgA among the months (Fig. 1A).

Analyzing the IgM production profile, the majority of samples were IgM positive and only one sample was IgM negative in the first month. As expected, IgM positivity declined in the following months (0.67 nm/0.42; 0.45 nm/0.60; 0.44 nm/0.52; 0.34 nm/0.53; 0.24 nm/0.58; 0.23 nm/0.50 and 0.26 nm/0.45, respectively to first until seventh month; mean/coefficient of variance), and the percentage of IgM positivity decreased gradually from 97% in the first month to 25% in the seventh month (Fig. 1B).

IgG positivity remained homogeneous for all 7 months. However, the mean OD, representing the semi-quantitative IgG values, was very high in the first month of analysis and perceptibly decreased over subsequent months (2.31 nm/0.18; 2.13 nm/0.31; 2.13 nm/0.21; 2.13 nm/0.19; 1.77 nm/0.28; 1.61 nm/0.32 and 2.01 nm/0.24, respectively to first until seventh month; mean/coefficient of variance), with a 1.35 times reduction in the fifth month and a 1.45 times reduction in the sixth month. Interestingly, the average of semi-quantitative IgG values increased again in the seventh month. It is worth noting that some patients were not IgG positive in the first, second, and sixth months (Fig. 1C).

IgE production was under the cut-off values for almost all the samples analyzed during the study. However, interestingly, of the only six IgE positive samples, two were positive in the first month, one in the third, and, finally, two in the seventh month (0.16 nm/0.31; 0.13 nm/0.19; 0.13 nm/0.24; 0.14 nm/0.16; 0.14 nm/0.23; 0.13 nm/0.15; 0.14 nm/0.24, respectively to first until seventh month; mean/coefficient of variance). Notably, the OD values for these six IgE positive patients were remarkably low when compared to the values obtained for IgM, IgG, and IgA, and only two samples, one of them in the first and the other in the third month, showed higher OD values compared to all the IgE positive samples (Fig. 1D).

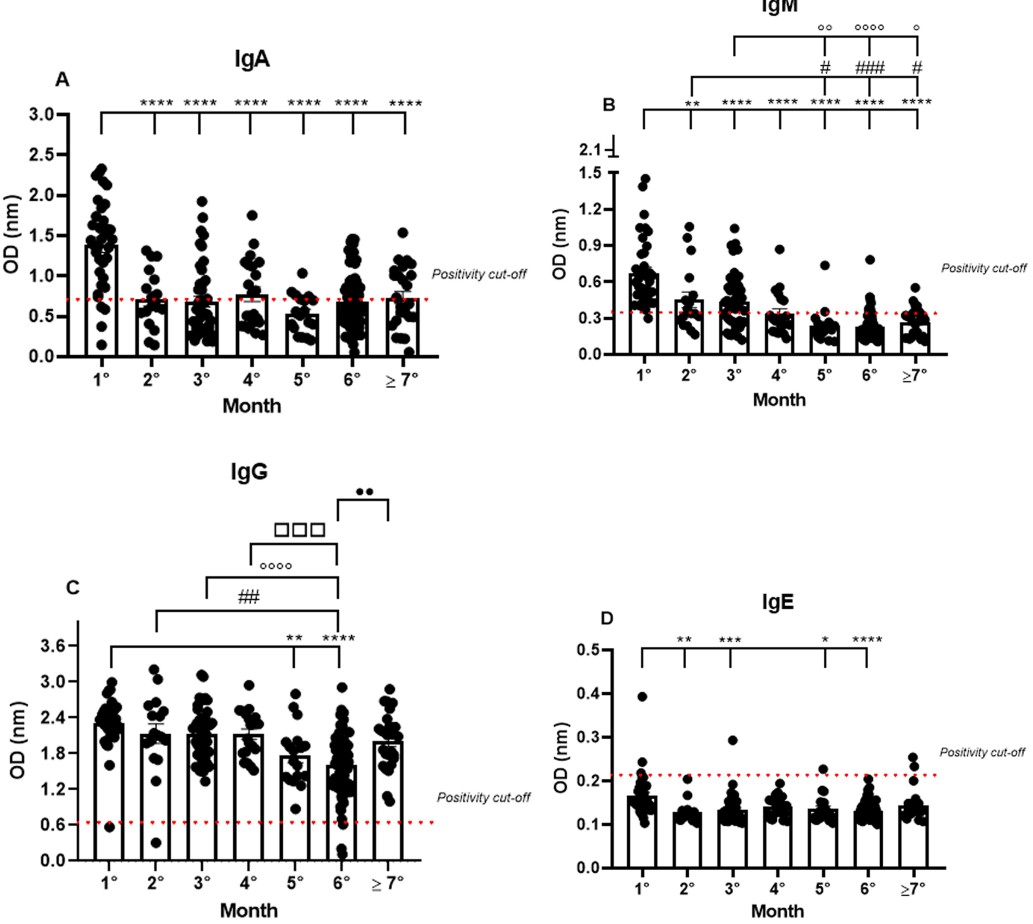

**Figure 1 The long-term quantification of IgA (A), IgM (B), IgG (C), and IgE (D) by S-UFRJ-ELISA assays.** The serum of 231 RT-qPCR-positive SARS-CoV-2 patients was measured for 7 months. The results were analyzed using the Kruskall–Wallis test followed by Dunn's *post hoc* test. The representative difference between: the first and subsequent months (*); the second and subsequent months (#); the third and subsequent months (o); the fourth and subsequent months (□), and the fifth and subsequent months (●). The symbols indicating statistical significance were $^{*,\#,o}p < 0.05$; $^{**,\#\#,oo,\bullet\bullet}p < 0.01$; $^{***,\#\#\#,\Box\Box\Box}p < 0.001$; $^{****,oooo}p < 0.0001$.

## Gender-related analyses of IgA, IgM, IgG, and IgE production against SARS-CoV-2

Subsequently, the data were analyzed comparing antibody production between genders over 7 months. The IgA positivity profile for the comparison of women and men patients was very similar for every month analyzed, and the previously noted decrease in IgA production was observed throughout the 7 months for both genders. Nonetheless, it is interesting to note the dispersion in OD in the first month (1.39 nm/0.39 and 1.38 nm/0.43 respectively, to women and men; mean/coefficient of variance), when women showed an elevated variation in IgA OD values in comparison to the values registered for men, which were higher and less dispersive (Fig. 2A).

IgM positivity did not show significant differences between the genders for most of the months. However, when comparing women and men from the fourth month after the

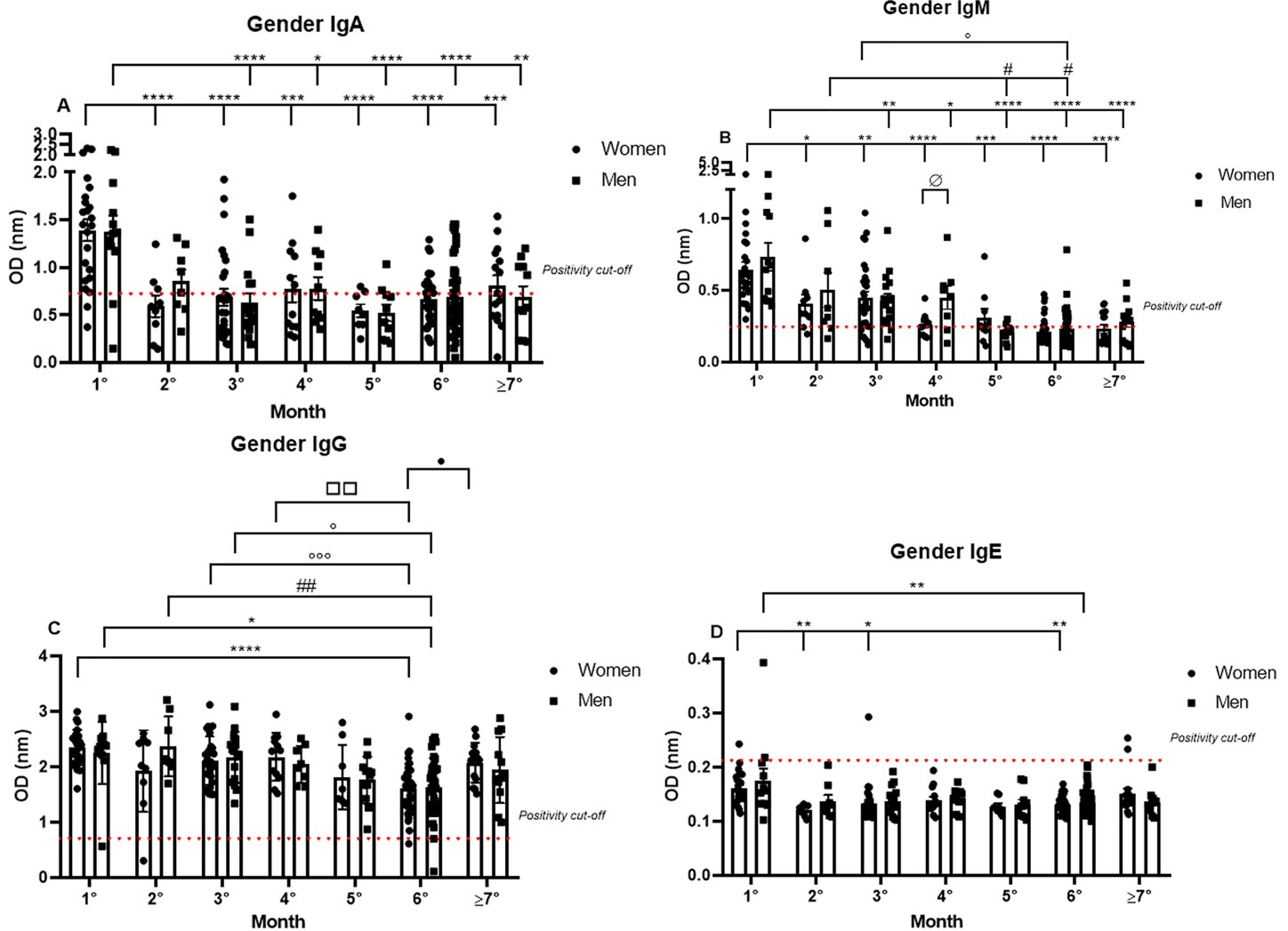

**Figure 2 The gender comparative analysis of IgA (A), IgM (B), IgG (C), and IgE (D) by S-UFRJ-ELISA assays.** The serum of 231 RT-qPCR-positive SARS-CoV-2 patients was measured for 7 months. The results were analyzed using the Mann–Whitney test. The representative difference-between: the first and subsequent months (*); the second and subsequent months (#); the third and subsequent months (o); the fourth and subsequent months (□); and for women and men on month four (ø). The significances were *,#,o,o,•$p < 0.05$; **,##,□□$p < 0.01$; ***,ooo$p < 0.001$; ****$p < 0.0001$.

onset of symptoms (0.26 nm/0.30 and 0.45 nm/0.51 respectively, to women and men; mean/coefficient of variance) there was a difference, and lower production was observed for women. Regarding the intersample variation, a difference between genders was noted, with female patients presenting more disperse and lower OD values than male patients (Fig. 2B).

The most homogeneous antibody production over the study period was observed for IgG, and no discrepant differences between men and women were observed at all, both for absolute production and for the individual dispersion in the semi-quantitative analysis (Fig. 2C).

Comparing IgE production between genders, it is possible to note that of the six IgE positive samples, four samples were from women and two were from men (Fig. 2D).

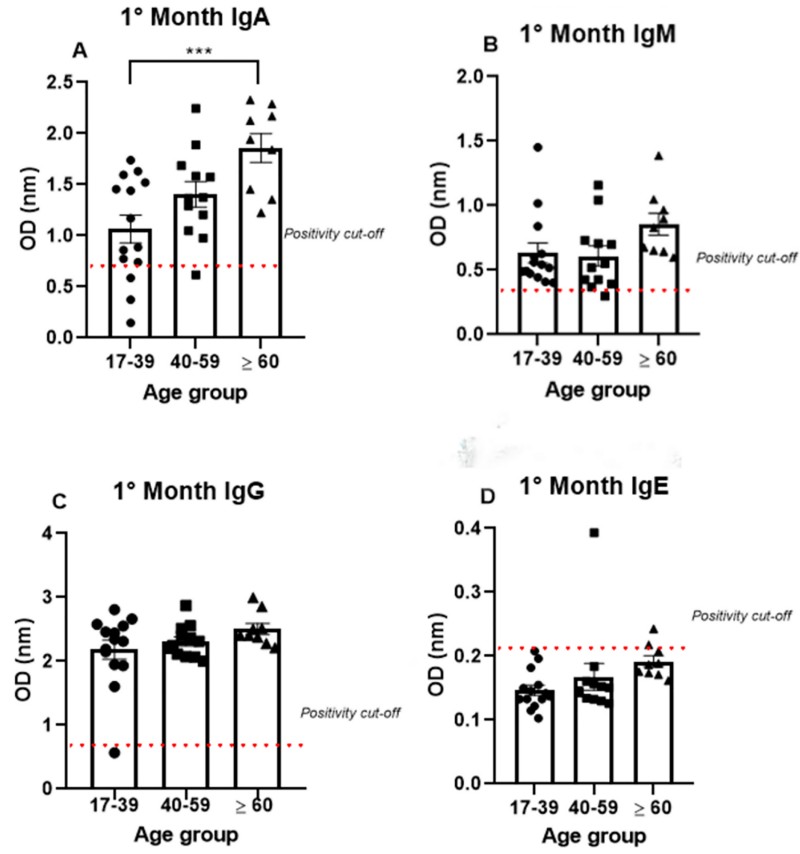

**Figure 3** **The age groups comparative analysis of IgA (A), IgM (B), IgG (C), and IgE (D) by S-UFRJ-ELISA assays.** Three age groups were separated into young adults, adults, and older adults. The serum of 34 RT-qPCR-positive mildly symptomatic SARS-CoV-2 patients from the first month was analyzed. Each dot represents an individual sample (17–39: *n* = 14; 40–59: *n* = 12; >60: *n* = 9, for age group). The results were analyzed using the Kruskall–Wallis test followed by Dunn's *post hoc* test. The representative difference between the age groups (*). The significance was ***$p < 0.001$.

## Age-related analyses of IgA, IgM, IgG, and IgE production against SARS-CoV-2

To analyze antibody production among the different age groups, the samples were divided into three groups (17–39, 40–59, and >60 years old), and the analysis were performed using only the data from the first month, because all the antibodies tested (IgA, IgM, IgG, and IgE) were produced at high levels mainly in the first month. IgA was produced after the onset of symptoms in all age groups (1.07 nm/0.48; 1.41 nm/0.31 and 1.86 nm/0.23, respectively, to 17–39, 40–59 and >60 years old mean/coefficient of variance). Interestingly, most samples that tested negative for IgG were from the young age group, while all the patients from the older group were IgA positive (Fig. 3). Notably, the average OD value was significantly different between the groups. These data indicate an increase in anti-SARS-CoV-2 IgA production with age, as the group containing older individuals produced almost 50% more IgA than the group containing the younger patients (Fig. 3A).

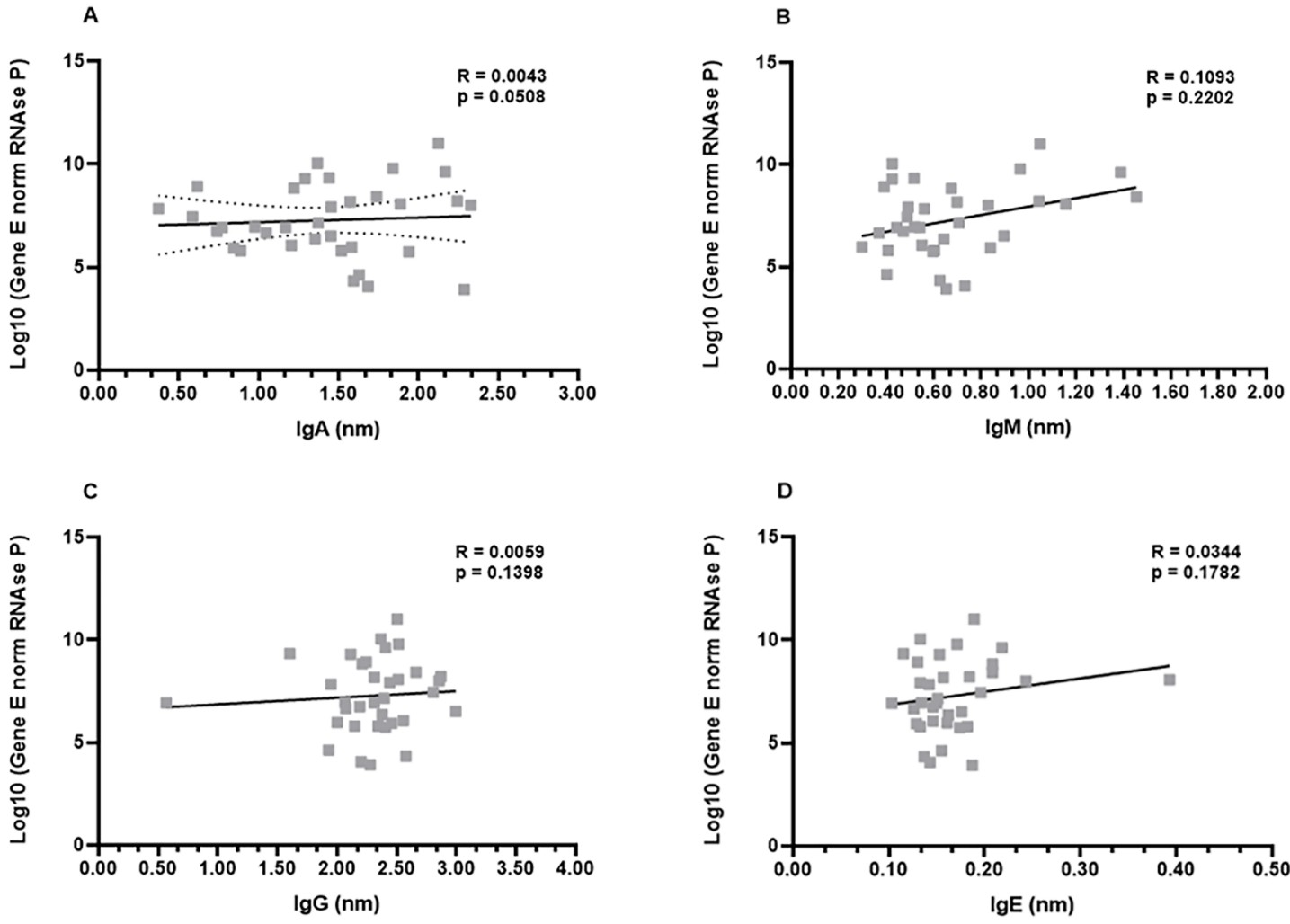

**Figure 4 Profiles of IgA (A), IgM (B), IgG (C), and IgE (D) and viral load.** A panel of 34 swab samples from PCR-positive mildly symptomatic SARS-CoV-2 patients was used to quantify viral load, which was presented as log(10) and calculated according to standard curve and numbers of copies/mL. The results were analyzed using Spearman's test.

Semi-quantitative production of both IgM and IgG presented a homogeneous distribution among the age groups. A subtle, non-significant increase in IgM production was observed in the older age group. The only IgM negative sample belonged to the 40–59 age group; while the only IgG negative patient was in the 17–39 age group (Figs. 3B, 3C).

The IgE analysis showed positivity in only one sample from the 40–59 age group and one sample from the ≥60 age group, as shown in the Fig. 3D.

## Analysis of SARS-CoV-2 viral load and IgA, IgM, IgG, and IgE production

In order to investigate a possible correlation between viral load and antibody production, the samples from the first month were subjected to the non-parametric Spearman's rank correlation analysis. The Spearman's coefficient ($\rho$) indicates a relationship between the production of each immunoglobulin and the viral load of the respective patient. This

analysis revealed that there was a low or inexistent positive correlation between viral loads and IgA, IgG, and IgE levels (ρ–values: 0.0508; 0.1398; 0.1782, for IgG, IgA, and IgE, respectively). Although not robust, a slightly increased positive correlation was observed for IgM production (ρ = 0.2202) (Figs. 4A–4D).

Next, an analysis was performed considering the genders (men and women) and viral load. Once again, the data revealed that there were no differences between men and women in the relationship between IgA, IgG, and IgE and viral load (Figs. S2A, S2C, S2D). The ρ coefficients considering the IgM levels once again presented a moderate correlation (ρ = 0.3261) for women. Men patients didn't show correlation (ρ = −0.0090) between viral load and IgM, possibly reflecting the fact that higher viral loads induce higher IgM production only in women (Fig. S2B).

Finally, the correlation analysis was performed taking into account the age groups of the COVID-19 patients. Again, no correlation was evident between the viral loads and levels of IgG, IgA, and IgE (Fig. S3). Interestingly, the ρ coefficients considering the IgM levels once again presented a statistical correlation (Fig. S3B).

## DISCUSSION

The current study followed the long-term dynamics of semi-quantitative serum levels of IgA, IgM, IgG, and IgE against S-glycoprotein from patients presenting mild COVID-19 symptoms over 7 months after infection. It is widely known that the antibody response is very important to eliminate viruses and to prevent new infections (Chvatal-Medina et al., 2021). Previous studies have reported high IgA and IgM production in the first month after the onset of COVID-19 symptoms, as well as a marked production decrease mainly in IgM and IgA after the first month, accompanied by continued production of IgG (Jin et al., 2020; Aita et al., 2020; Sterlin et al., 2021). This study presents a different scenario, which IgA production persists in more than 60% of patients during the second and third months after the onset of symptoms for patients with mild symptoms. Regarding IgG, its prolonged production was observed here as in previous studies that have shown the involvement of this antibody class to prevent reinfection. Indeed, one previous study revealed that detectable IgG levels persisted for more than 2 years in SARS-CoV and MERS-CoV infected patients (Zhou et al., 2021). The present study showed continued IgG production over the 7 months, and a slight decrease was observed for the last months indicating that from fifth month after infection, the IgG production are not produced for long periods in high quantities. In this study, some positive samples of IgM was observed after the first month, generally, in a clinical diagnosis, a high level of IgM antibody is considered as an indicator of acute infection or recent reinfection. However, in this study, the patients did not report symptoms of COVID-19 or did not perform a RT-qPCR more the once time during the seventh months analyzed, indicating that they may not had a reinfection. However, this hypothesis cannot be discarded, because some patients are asymptomatic (Pozzobon et al., 2022). Another two hypothesis which could explain positive samples of IgM after first month is concerning a persistent SARS-CoV-2 infection in some patients, culminating in a persistent positive production of IgM antibodies as reported by some authors (Choudhary et al., 2022). Finally, Fanfan et al. (2021) reported,

in patients infected once time by SARS-CoV-2, who were analyzed IgM titers and followed up for 1 year, a persistent positive IgM production (12.80%). Then, it is not reliable and proper to assist clinical diagnosis of infection only by IgM detecting results. Moreover, the reason underlying such a long duration of SARS-CoV-2 IgM in some patients still unclear and needs additional study.

Taking gender into account, previous research identified that there were more women with high levels of IgG relative to men with a severe COVID-19 status. Here, in this study analyzing mild cases, no difference between genders was observed for IgG and, only a subtle difference was observed for IgM in the fourth month, indicating that severity could to influence in the dynamic of the humoral response among genders; however, with respect to mild cases gender may not have an influence (*Zeng et al., 2020*).

Interestingly, in this study, taking account of viral load, IgM, and gender, revealed that only women produced IgM according to their viral load, even though, men did not show correlation between IgM and viral load, they presented the high IgM production comparing to women. This fact could be explained by a lower viral load that women were exposure during this study. In another study performed in Macae, which analyzed only viral load and gender, and did not consider IgM titers, it was observed that, there were a higher percentage of men at working efforts compared to women that showed high viral loads. Men make up the main workforce in the essential activities of offshore companies from Macaé involved in oil exploration of the Brazilian coast, which were not interrupted during the quarantine. Taken together, these results suggest that actively working men were more exposed to virus, showing a high production of IgM than women.

The viral load analysis comparing age and IgA, IgM, IgG, and IgE titers showed a positive correlation only between IgM production and viral load in the older age group (>60 years old). IgM normally occurs in high quantities during the first days of an acute infection, playing an important role in initiating the host humoral defense mechanism. Due to its early role in the humoral immune response, it is conceivable that IgM levels exhibit a positive correlation with viral loads, as occurs with the older COVID-19 patients in this study (*Yang et al., 2021*). In contrast, the younger patients do not exhibit this correlation, suggesting that initial humoral immune responses to COVID-19 differ depending on the age, but not on the gender of the patient, and, it may be that an immune cellular response could be more activated being more effective than a humoral response to eliminate the SARS-CoV-2 virus in the younger group. Nonetheless, it is important to understand the pathophysiologic basis underlying the different disease manifestations and outcomes of SARS-CoV-2 infection in patients from all age groups.

Our findings suggest that the differences in the clinical manifestations of COVID-19 in different age groups might be partly due to age-related immune responses. Here, we observed significantly increased titers not for IgG but for IgA levels, when comparing the age groups. Thus, the older (>60 years old) patients showed a higher level of IgA than the younger ones. Similarly to our results, comparing IgG titers between ages revealed the same pattern of antibody production (*Yang et al., 2021*).

Nevertheless, the exact mechanisms inherent in the different SARS-CoV-2 immune responses based on age remain unclear. It may be that adolescents and young adults have

attenuated immune responses that culminate in more tolerance to the virus. On the other hand, some other studies have suggested a more active innate immune response in adolescents and young adults than in older adults, which culminates in efficient virus elimination and low levels of antibodies (*Elahi, 2020*; *Zimmermann & Curtis, 2020*; *Castagnoli et al., 2020*).

In addition, some research has proposed that trained immunity may develop a function and that innate immune memory generated by other vaccines may confer a nonspecific protective effect against SARS-CoV-2 (*Fidel & Noverr, 2020*). Interestingly, our data revealed that IgA and IgM production does not present a substantial correlation to viral load, indicating that viral load does not induce a high humoral response to eliminate the virus and that the innate response could perhaps develop a key role in SARS-CoV-2 infection. Thus, these data indicate that more investigations are needed to better understand the age dynamic of the cellular and humoral immune response.

Some other previous studies exploring serological assays directed at SARS-CoV-2, but analyzing only the IgG and total antibody titers, observed higher levels in the serum of severe patients than in non-severe ones in the older age group, which was associated with the establishment of a critical inflammatory process and hospitalization (*Yang et al., 2021*). These data correlate an elevated humoral response with the presence of previous chronic diseases in older patients.

*Zervou et al. (2021)* conducted an study analyzing IgA to SARS-CoV-2 and concluded that IgA levels were highest in patients with a severe and critical illness. Here, the elevated IgA production was observed in a mild symptomatic older age group and could be associated with an increase in comorbidities, such as obesity, diabetes, and chronic cardiac diseases, among others (*Zervou et al., 2021*). These comorbidities, especially obesity, were described to have a high baseline of pro-inflammatory cytokines and SARS-CoV-2 infection could have a stimulatory effect on humoral response, which culminates here with the observation of high IgA production in these patients (*Ejaz et al., 2020*; *Kulcsar et al., 2019*). It may be that the humoral responses in the older group could be more activated because some comorbidities such as diabetes and chronic cardiac diseases are prevalent in this group, which are known for their influences on the cellular immune responses, including impairing phagocytic cell capacities and inducing high levels of ACE2 receptors (*Rao, Lau & So, 2020*).

Analyzing this hypothesis, the comorbidities data in the age groups revealed that the percentage of obesity is high in all age groups (19.6% in adolescents and young adults, 45.5% in adults, and 31.2% in seniors) and this probably was not the main factor that influenced the high IgA production. However, comparing chronic cardiac diseases and diabetes and the age groups, it was possible to note that these diseases were more prevalent in the older group than in other age groups and may contribute to the high levels of IgA observed here. Lastly, more investigation is currently needed to better understand the age-related dynamics of the cellular and humoral immune responses elicited against SARS-CoV-2.

Regarding IgE, the most common responses to viruses are mediated by IgM, IgG, and IgA. Non expected, type-2 immunity, common for anti-helminth defense through IgE, was

observed in severe COVID-19 patients, who showed increased levels of IgE during the first 10 days of disease (*Lucas et al., 2020*). Here, IgE production was also analyzed during the 7 months after the onset of symptoms in mild COVID-19 patients. Our data revealed that only six patients were IgE positive, showing slight production of these molecules. Indicating that more researches are needed to disclose the function of the IgE in the dynamic of antibodies production of the COVID-19 disease.

We recognize that there are some limitations to this study. We used 231 serum samples from RT-qPCR-confirmed COVID-19 patients; however, serum samples were not available every month for each patient during the 7 months. Nevertheless, we believe this study provides valuable information regarding the serological analysis of COVID-19 considering the parameters of gender, age, viral load, comorbidities, and duration of responses for IgM, IgG, IgA, and IgE antibodies in mildly symptomatic patients.

In summary, our study provides additional evidence to help guide the use of serologic testing in the diagnosis and management of COVID-19 infection. Additionally, the data presented here can help in a correlation between s-antibodies to SARS-Cov-2 and the emerging knowledge for the development of immunotherapeutics and vaccines. However, further studies of antibody persistence in SARS-CoV-2, reinfection and long COVID are warranted.

## ACKNOWLEDGEMENTS

The authors gratefully acknowledge the donation of trimeric spike protein of SARS-COV-2 by the Cell Culture Engineering Laboratory of COPPE/UFRJ—Federal University of Rio de Janeiro.

### Funding

This research was supported by the Justiça Federal de Macaé (JFM), the Justiça Federal de Itaperuna (JFI), and the Fundação Carlos Chagas Filho de Amparo à Pesquisa do Estado do Rio de Janeiro (FAPERJ), through grant numbers E-26/210.176/2020/ and E-26/210.822/2021. The Institute of Biodiversity and Sustainability (NUPEM) of the Federal University of Rio de Janeiro (UFRJ) received support from the Ministério Público Federal, the Ministério Público do Trabalho, the Confederação Nacional das Cooperativas Médicas (UNIMED), Irmandade São João Batista, the Prefeitura Municipal de Macaé, the Tribunal Regional Federal da 4ª Região (TRF-4), the Associação dos Docentes da UFRJ (ADUFRJ), and donors that contributed to this research. The complete list of donors is available at http://transparencia.coppetec.ufrj.br/pesquisa-covid19-macae.php. The funders had no role in study design, data collection and analysis, decision to publish, or preparation of the manuscript.

### Grant Disclosures

The following grant information was disclosed by the authors:
Justiça Federal de Macaé (JFM).

Justiça Federal de Itaperuna (JFI).
Fundação Carlos Chagas Filho de Amparo à Pesquisa do Estado do Rio de Janeiro
(FAPERJ): E-26/210.176/2020/ and E-26/210.822/2021.
Ministério Público Federal.
Ministério Público do Trabalho.
Confederação Nacional das Cooperativas Médicas (UNIMED).
Irmandade São João Batista.
Prefeitura Municipal de Macaé.
Tribunal Regional Federal da 4ª Região (TRF-4).
Associação dos Docentes da UFRJ (ADUFRJ).

## Competing Interests

Rodrigo Nunes Da Fonseca is an Academic Editor for PeerJ.

## Author Contributions

- Graziele Fonseca de Sousa performed the experiments, analyzed the data, prepared figures and/or tables, authored or reviewed drafts of the article, and approved the final draft.
- Thuany da Silva Nogueira performed the experiments, analyzed the data, prepared figures and/or tables, authored or reviewed drafts of the article, and approved the final draft.
- Lana Soares de Sales performed the experiments, analyzed the data, prepared figures and/or tables, investigation; data curation, and approved the final draft.
- Fernanda Ferreira Maissner performed the experiments, prepared figures and/or tables, and approved the final draft.
- Odara Araújo de Oliveira performed the experiments, prepared figures and/or tables, and approved the final draft.
- Hellade Lopes Rangel performed the experiments, analyzed the data, prepared figures and/or tables, investigation; data curation, and approved the final draft.
- Daniele das Graças dos Santos performed the experiments, analyzed the data, prepared figures and/or tables, investigation; data curation, and approved the final draft.
- Rodrigo Nunes-da-Fonseca analyzed the data, authored or reviewed drafts of the article, resources; Funding Acquisition, and approved the final draft.
- Jackson de Souza-Menezes analyzed the data, authored or reviewed drafts of the article, review and editing, and approved the final draft.
- Jose Luciano Nepomuceno-Silva performed the experiments, analyzed the data, authored or reviewed drafts of the article, and approved the final draft.
- Flávia Borges Mury performed the experiments, analyzed the data, authored or reviewed drafts of the article, review and editing, and approved the final draft.
- Raquel de Souza Gestinari performed the experiments, analyzed the data, authored or reviewed drafts of the article, review and editing, and approved the final draft.
- Amilcar Tanuri analyzed the data, authored or reviewed drafts of the article, resources; Funding Acquisition, and approved the final draft.

- Orlando da Costa Ferreira Jr conceived and designed the experiments, analyzed the data, authored or reviewed drafts of the article, resources; Funding Acquisition, and approved the final draft.
- Cintia Monteiro-de-Barros conceived and designed the experiments, performed the experiments, analyzed the data, authored or reviewed drafts of the article, resources; Funding Acquisition, and approved the final draft.

## Human Ethics

The following information was supplied relating to ethical approvals (*i.e.*, approving body and any reference numbers):

The study was approved by the Comitê de Ética em Pesquisa NUPEM-UFRJ Institute (Research Ethics Committee, Brazilian Ministry of Health: approval number 32868720.4.0000.5699).

## Data Availability

The raw measurements are available in the Supplemental File.

## Supplemental Information

Supplemental information for this article can be found online at http://dx.doi.org/10.7717/peerj.14547#supplemental-information.

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
