# Peer review of "The long-term dynamics of serum antibodies against SARS-CoV-2"

_PeerJ, doi:10.7717/peerj.14547_

## Round 0.1 · original submission · Minor Revisions

When resubmitting the revised version of the manuscript you should include a point-by-point reply to the reviewer comments. Further, please use tracked changes in the revised version. In case you need more than 21 days for revision, it is no problem to extend the deadline.

Reviewer 1 ·

Basic reporting

The manuscript is overall well-written with the conclusions clearly presented. Some of the figures (especially figure 1) are a bit blurry and the axis labels could be hard to read.
The introduction provided sufficient background and relevant studies were cited to compare findings.

Experimental design

The research questions were well-defined. One of the issues with the study, which the authors briefly acknowledged at the very end of their discussion, was that longitudinal data from the same cohort of patients could not be obtained for the entire duration of the study. This issue resulted in small group sizes in some of the analyses in the study that involved subsetting patients by gender and age, with conclusions being drawn from samples from about 10 patients. Because of these low numbers, the authors would need to address the potential effect of random sampling on their data.

A similar concern that needs to be addressed is regarding the viral load measurements. According to the raw data, these measurements were taken anywhere between 1 to 12 days post symptom onset, which is a large range. Given the kinetics of viral clearance, the authors should rule out the effect of time when samples were collected and viral load (i.e. is viral load higher at a certain day post infection/symptom onset) prior to testing the latter’s correlation with gender and age.

A minor question is the potential effect of antigen re-encounter on the antibody titer. Given the sample were collected between June 2020 to January 2021, it is possible that some of the patients could have repeated exposure to the virus. Could the bimodal distribution (or outliers) in some of the data the authors observed be a result of the uptick in antibody titer following antigen re-encounter?

Validity of the findings

The authors might consider statistical tests comparing variances in cases where they stated an increased variability within groups (for example in figure 3).

The conclusion that “IgA positivity gradually decreased after the first month, but started to increase again in the seventh month” (line44) in the abstract was not supported by the data presented. Similarly, conclusions were slightly overdrawn in the discussion section. For example, lines 297-299 state “taking account of viral load, IgM, and gender, revealed that both genders (men and women) produced IgM according to their viral load, which women produced less IgM than men” despite in both cases the correlation coefficients were less than 0.2 with p values greater than 0.05.

Reviewer 2 ·

Basic reporting

No comments

Experimental design

- Methods: How was the cut offs for Igs were decided. Elaborate in methods.

- Pg 13: Line 183-190: It would be better if authors also showed the values (% change in IgA) as bar graphs in Figure 1A. Write in the figure legend, what each dot (filled circle) represents?

- How are the sample selection done for performing the ELISA. Do mention the selection criteria in figure legends.

Validity of the findings

This study adds to the present knowledge in the field. Manuscript is well written and the results are appropriately discussed. It will be better if the authors add some speculations for some future studies and how it can help in developing understanding towards SARS-CoV-2 immunity.

Additional comments

- Authors should use either R or p for denoting correlation coefficient. The symbol is different in text and figure.
- Line 372-75: Reframe the sentence to remove grammatical error.

Reviewer 3 ·

Basic reporting

The manuscript by Sousa GF et.al has investigated the serological profiles of anti-spike antibodies post SARS-COV2 infection for 7 months, and looked at the dependance of serology on age, gender, and viral loads. The manuscript is well-written in unambiguous and professional English. There are minor typographical errors in the main manuscript and the supplemental materials which can be easily fixed on proofreading. The introduction and background behind the study is nicely explained, well-researched, structured with latest studies. However, how the aim of the study has been introduced suddenly, without linking it to the knowledge gap left by the previous studies (line 108-111). I suggest the authors to rephrase their aim so that it syncs with the introduction and background. The structure of the manuscript conforms to PeerJ guidelines.

Experimental design

The experimental design of the study is scientifically sound and mostly well-executed. The materials and methods are lucidly explained for future replications. However, there are some minor and major suggestions that I have enumerated below:
• What was the day post-symptom and post-positive PCR tests for first draw of each subject? Did all the 231 samples came from unique subjects? Or were there longitudinal follow-ups of same subjects? The manuscript is not clear on this.
• For Table 1, it will be more informative if the comorbidities can be split up between the two genders.
• Figure 1B: The Y axis upper segment scale markings are overlapped, please check it
• One general comment is that the authors can consider giving the n values in the figure legends, and not in the x-axis. This will make the figures less cluttered.
• The total number of subjects in 1st month adds up to 34 as per Figure 3. However, the supplementary information shows PCR swab was done from 35 patients. Please clarify why 1 patient was not assayed for serology?
• What was the average time-point for 1st month draw? It will be interesting to see if there is any serological difference in the 1st month itself, we the authors compares acute phase of infection (day7-10) and late phase of infection at around day20-day30.
• I would also suggest that the authors show serological time-course for IgG, IgM, IgA and IgE separately for different age groups. It will be interesting to see how the anti-spike isotypes vary amongst different age groups.

Validity of the findings

The authors have provided appropriate conclusions for all the figures. The study has been based on human samples, which further strengthens the study and proves its importance to the field. However, there is need of more data/clarification in some cases, which I have listed below:
• One question is whether the individuals included in the cohort had any kind of immunosuppressive therapy? The serological profiles can be affected by immunosuppressive drugs.
• Was there a difference in viral loads among different age groups, and between the genders? There are reports of IgA levels being positively correlated with disease severity (https://doi.org/10.1002/jmv.27058), and Figure 3A shows increased IgA in the old age group (above 60years).

Additional comments

None

---

## Round 0.2 · accepted · Accept

I herewith confirm that you have addressed all the reviewers' comments and, thus, your manuscript is suitable for publication.

Reviewer 1 ·

Basic reporting

The authors have addressed reviewers' comments by improving figure legibility and adding a transitional introduction paragraph.

Experimental design

The authors have sufficiently addressed all of the reviewers’ comments including adding clarifications of experimental methods and patient cohort information and performing variance analysis.

Validity of the findings

The authors have addressed reviewers' concerns including editing sentences where conclusions were overdrawn or unclear.

Additional comments

As the authors have sufficiently addressed comments from reviewers, I believe the manuscript is now ready for publication.

Reviewer 2 ·

Basic reporting

No further comments

Experimental design

No further comments. Authors have modified the manuscript as per my recommendations.

Validity of the findings

No further comments

Additional comments

No further comments

Reviewer 3 ·

Basic reporting

All the necessary concerns have been addressed by the authors, and there is no more concerns.

Experimental design

The suggested edits have been incorporated by the authors and no concern remains.

Validity of the findings

All the concerns have been satisfactorily answered.

Additional comments

None